

# Osteology of a forelimb of an aetosaur *Stagonolepis olenkae* (Archosauria: Pseudosuchia: Aetosauria) from the Krasiejów locality in Poland and its probable adaptations for a scratch-digging behavior

Dawid Dróżdż

Institute of Paleobiology, Polish Academy of Sciences, Warsaw, Poland

## ABSTRACT

Aetosaurs are armored basal archosaurs that played a significant role in land ecosystems during the Late Triassic (237–201 Ma). The polish species *Stagonolepis olenkae* Sulej, 2010 described from the Krasiejów locality (southern Poland) is one of the oldest known representatives of the group. Abundant and well-preserved material, including partially articulated specimens, allows a detailed description of the forelimbs in this species. The forelimbs of *S. olenkae* are the most similar to that of large aetosaurs like *Desmatosuchus smalli*, *Desmatosuchus spurensis*, *Longosuchus meadei*, *Typothorax coccinarum* or *Stagonolepis robertsoni*. Several characters recognized in the forelimbs of *S. olenkae* suggest its adaptation for scratch-digging. The most salient of these features are: short forearm, carpus, and hands, with the radius shorter than the humerus, carpus and manus shorter than the radius (excluding terminal phalanges); a prominent deltopectoral crest that extends distally on the humerus and a wide prominent entepicondyle, a long olecranon process with well-marked attachment of triceps muscle; hooked, laterally compressed, claw-like terminal phalanges with ornamentation of small pits (indicative of well-developed keratin sheaths). *S. olenkae* might have used its robust forelimbs to break through the compacted soil with its claws and proceed to dig in search of food in softened substrate with the shovel-like expansion at the tip of its snout. The entire forelimb of *S. olenkae* is covered by osteoderms, including the dorsal surface of the hand, which is unusual among aetosaurs and have not been noted for any species up to date.

# INTRODUCTION

Aetosaurs are heavily armored, quadrupedal basal archosaurs cladistically nested within Pseudosuchia, the crocodile lineage of archosaurs (e.g., *Brusatte et al., 2010*; *Nesbitt, 2011*; *Desojo, Ezcurra & Kischlat, 2012*; *Desojo et al., 2013*). They are medium to large sized animals (one to six m long) with semi-erect to erect gait (*Parrish, 1986*; *Desojo & Báez, 2005*;

Corresponding author
Dawid Dróżdż,
dawid.drozdz@twarda.pan.pl

*Desojo & Vizcaíno, 2009*; *Heckert et al., 2010*; *Padian, Li & Pchelnikova, 2010*; *Desojo et al., 2013*). Their most characteristic feature is the suit of dermal armor composed of rectangular, plate-like osteoderms, which cover the dorsal and partially the lateral surfaces of their bodies, and in more heavily armored species also the belly, ventral surfaces of the tail and the limbs (e.g., *Walker, 1961*; *Heckert & Lucas, 2000*; *Desojo et al., 2013*). Presumably, the aetosaurs are considered to be omnivores but their exact mode of life is not clear yet. In most species (with the exception of *A. ferratus*) the triangular skull is equipped with a shovel-like expansion at the tip of tapering snout (e.g., *Desojo & Vizcaíno, 2009*; *Sulej, 2010*; *Desojo et al., 2013*). They probably used this shovel-like expansion to dig their food out of the ground (e.g., *Walker, 1961*; *Sulej, 2010*; *Desojo et al., 2013*), likely utilizing limbs in the process (*Heckert et al., 2010*). Aetosaur fossil remains are restricted in occurrence to the continental Upper Triassic (Carnian–Rhaetian) (e.g., *Heckert & Lucas, 2000*; *Desojo et al., 2013*). They are known from several localities in Europe, India, Africa, and both Americas, which makes them one of the most widespread groups among pseudosuchians in the Late Triassic (e.g., *Desojo et al., 2013*).

In Poland, aetosaurs have been reported from two localities in southern part of the country: Krasiejów and Poręba. The material described from Poręba is restricted to few osteoderms and a vertebra, which hampers further studies and attempts of taxonomical affiliations (*Sulej, Niedźwiedzki & Bronowicz, 2012*). In contrast, the aetosaur collection from Krasiejów is very rich and provides several cranial and numerous postcranial specimens of different size and probably different ontogenetic age. Although the material is so abundant, there is some controversy over its taxonomical affiliation. In the detailed description of the skull by *Sulej (2010)*, the aetosaur from Krasiejów has been recognized as a new species, *Stagonolepis olenkae* Sulej, 2010, similar to *Stagonolepis robertsoni* Agassiz, 1844 from Elgin, Scotland. Further work by *Książkiewicz (2014)* on the postcranium of the Krasiejów species also supported this thesis. However, *Antczak (2015)* suggested synonymy of *S. olenkae* and *S. robertsoni* based on new skull material. He argued that previously recognized differences represent intraspecific variations or sexual dimorphism. Quite different results were shown by the cladistic analysis by *Parker (2016)* and *Parker (2018)*, also based on the cranial material. It placed *S. olenkae* and *S. robertsoni* on two distant branches of the tree, implying that they may even represent distinct genera. I also recognized some small differences between those two species, but after personal examination of material assigned to both species I have to admit that differences in the postcranium of those two species are minor in general and difficult to interpret, mostly due to small sample size and incompleteness of the preserved elements in Scottish material and its likely younger ontogenetic age, as suggested by smaller sizes of the specimens and some other indicatives (e.g., lack of fusion between the elements of the axial skeleton or pectoral girdle). Until some more comprehensive taxonomic and ontogenetic study is performed and the definitive answer is found, I follow the division into two species proposed by *Sulej (2010)* and I assign the aetosaur forelimb elements described in this study to *S. olenkae.*

In general, fossils of aetosaur forelimbs are less abundant than those of the hind limbs. In spite of that, the long bones of the forelimbs are recognized and well-described in several species (e.g., *Sawin, 1947*; *Walker, 1961*; *Bonaparte, 1971*; *Lucas, Heckert & Hunt,*

*2002*; *Schoch, 2007*; *Heckert et al., 2010*; *Roberto-Da-Silva et al., 2014*). Much less is known about aetosaurs' manus and carpus. In fact, those parts are described in detail only for *S. robertsoni*, *Longosuchus meadei*, *Typothorax coccinarum* and recently *S. olenkae* (*Sawin, 1947*; *Walker, 1961*; *Lucas & Heckert, 2011*; *Książkiewicz, 2014*; this study), but associated metacarpals with elements of carpus are known also for *A. ferratus* (*Schoch, 2007*). The material of *S. olenkae* currently consists of a several isolated humeri, radii and ulnae, and few specimens with associated forearm, hand and carpal elements. Two of those specimens are illustrated by *Książkiewicz (2014)* in his doctoral thesis, and four are described and illustrated in this study.

Based on the literature, large and middle size aetosaurs (over two m in length), like *L. meadei*, *S. olenkae*, *S. robertsoni*, and *T. coccinarum*, have forelimbs more similar to each other than to that the smaller aetosaurs (below one m in length), like *A. ferratus* and *Polesinesuchus aurelioi* (*Sawin, 1947*; *Walker, 1961*; *Long & Murry, 1995*; *Schoch, 2007*; *Heckert et al., 2010*; *Roberto-Da-Silva et al., 2014*). However, in all species the forelimbs are shorter than the hind limbs, robust and strongly-built, with the humerus longer than the forearm bones, and the radius and ulna of comparable length (e.g., *Sawin, 1947*; *Walker, 1961*; *Heckert et al., 2010*; *Desojo et al., 2013*). The humerus has a wide, well-defined head and a prominent deltopectoral crest (e.g., *Heckert et al., 2010*; *Desojo et al., 2013*). The ulna and radius are straight and the olecranon process of the ulna is usually high (e.g., *Desojo et al., 2013*). In the hand of known species, metacarpals and phalanges are short and wide, and the digits are finished with claw-shaped ungals (e.g., *Sawin, 1947*; *Walker, 1961*; *Lucas & Heckert, 2011*; *Desojo et al., 2013*). The probable phalangeal formula is 2-3-4-5-3 (*Walker, 1961*; *Desojo et al., 2013*) with the fifth digit much smaller than the others (*Lucas & Heckert, 2011*). The carpus consists of few (probably four) bony components (*Sawin, 1947*; this study). The most characteristic carpal element that distinguishes aetosaurs' carpus from that of other pseudosuchians is a large bone adjacent to radius, probably the fused radiale and intermedium (*Sawin, 1947*; *Walker, 1961*; *Schoch, 2007*; *Książkiewicz, 2014*, this study).

Probable use of the forelimbs for digging was already suggested for *S. robertsoni* by *Walker (1961)*, based on the presence of short manus and an excavation below the proximal end of the ulna. Adaptations for digging has been more comprehensively discussed by *Heckert et al. (2010)* in respect to *T. coccinarum*, known from almost complete and articulated skeletons. *T. coccinarum* possesses five of eleven characters that, according to *Hildebrand (1988)*, indicate a digging behavior in modern vertebrates, namely: a low brachial index (radius shorter than humerus), (1) a prominent deltopectoral crest that extends distally on the humerus, (2) a wide entepicondyle, (3) short and wide metacarpals, (4) short and wide phalanges (*Heckert et al., 2010*). In addition they pointed out that the compact and strongly articulated foot with a large, curved and laterally compressed ungals could have been used for scratch-digging, as it was suggested for the rhynchosaur *Hyperodapedon* (*Heckert et al., 2010*; *Benton, 1983*). Based on the presence of such digging adaptations in *T. coccinarum*, it is suggested that it might have used its limbs to unearth roots or burrowing invertebrates (*Heckert et al., 2010*; *Desojo et al., 2013*). The digging features mentioned for *T. coccinarum* are also present in the forelimbs of *S. olenkae*, but the latter also has other characters that

are indicative more specifically for scratch-digging, according to Hildebrand (*1983*, *1988*, *Heckert et al., 2010*).

The postcranial skeleton of *S. olenkae*, including forelimbs, has been described by *Książkiewicz (2014)* in his doctoral thesis. However, the research conducted for my paper was done independently of his studies, and is based on different material (with exception of hand elements of the specimen ZPAL AbIII/2071, which were only mentioned in his thesis but have not been illustrated). I have personally examined the material used by Książkiewicz, but since he wishes to publish his thesis soon, I will make only necessary references to his work, which have an impact on the subject of my paper. My work contains more detailed description of the *S. olenkae* forelimbs and discuss the probable scratch-digging behavior in this species, which is not discussed by *Książkiewicz (2014)*.

## GEOLOGICAL SETTING

Krasiejów is located at the southern-eastern edge of the fore-Sudetic Homocline in Opole Voivodeship (Upper Silesia), SW Poland. This was the first of the series of Late Triassic localities in Silesia containing bones of large vertebrates discovered near the end of the 1990s, supplemented later by the Lisowice, Poręba, Woźniki, and Marciszów localities in the beginning of the 2000s (*Sulej et al., 2011*; *Dzik et al., 2000*; *Dzik, Niedźwiedzki & Sulej, 2008*; *Dzik, 2001*; *Budziszewska-Karwowska, Bujok & Sadlok, 2010*; *Niedźwiedzki, Sulej & Dzik, 2012*; *Sulej, Niedźwiedzki & Bronowicz, 2012*; *Szczygielski & Sulej, 2016*; *Szczygielski, 2017*). The outcrop is located in an unoperational clay pit. The exact age of the beds exposed at Krasiejów is not certain. Originally, a late Carnian age was proposed, based on biostratigraphic evidence (*Dzik et al., 2000*; *Dzik & Sulej, 2007*; *Dzik & Sulej, 2016*; *Zatoń, Piechota & Sienkiewicz, 2005*; *Kozur & Weems, 2010*) but the lithostratigraphic correlations suggest the Norian age (*Szulc, Racki & Jewuła, 2015*).

Rocks deposited in Krasiejów consist mostly of red to red-brown or gray siltstones and mudstones. Alternating series of red and gray rocks indicate a seasonal climate with dry (red) and wet (gray) periods (*Szulc, 2005*; *Gruszka & Zieliński, 2008*). Lenses of fine-grained sandstone and calcareous concretions (often containing bones) as well as several paleosol horizons (*Szulc, 2005*) occur in the section. Below the level of the outcrop the presence of gypsum was reported (*Szulc, 2005*).

The Krasiejów locality is one of the largest accumulations of fossil vertebrates in central Europe. Specimens gathered from this locality number in the thousands. In addition to *S. olenkae*, the Krasiejów fauna includes the temnospondyls *Metoposaurus krasiejowensis* and *Cyclotosaurus intermedius*, basal archosaurs, including a rauisuchian *Polonosuchus silesiacus* and a phytosaur *Paleorhinus* (=*Parasuchus*) sp., the dinosauromorph *Silesaurus opolensis*, the gliding protorosaurid *Ozimek volans*, fishes, and invertebrates (*Dzik et al., 2000*; *Dzik, 2001*; *Dzik, 2003a*; *Dzik, 2003b*; *Dzik, 2008*; *Sulej, 2002*; *Sulej, 2005*; *Sulej, 2007*; *Sulej, 2010*; *Sulej & Mayer, 2005*; *Dzik & Sulej, 2007*; *Dzik & Sulej, 2016*; *Brusatte et al., 2009*; *Piechowski & Dzik, 2010*; *Skrzycki, 2015*). Plant remains are rare (*Dzik et al., 2000*; *Dzik, 2003b*; *Dzik & Sulej, 2007*; *Pacyna, 2014*).

The majority of vertebrate fossils occur in two bone-bearing horizons (e.g., *Dzik et al., 2000*; *Dzik & Sulej, 2007*; *Szulc, 2007*; *Bodzioch & Kowal-Linka, 2012*). The lower bone

horizon is about 1 m thick, with a grey bed (about 0.3 m thick) at the bottom that is distinct from the red paleosol series below it, and the red bed in its upper part covered mostly by a layer of calcareous grainstone and fluvial variegated sediments above. Most of the vertebrate fossils are preserved at the boundary of the red and grey sediments within this bone horizon. Horizontal layering of sediments indicates its deposition in open standing water (*Dzik et al., 2000*; *Dzik & Sulej, 2007*; *Gruszka & Zieliński, 2008*). In this bone level, fossils of aquatic vertebrates dominate: *Metoposaurus krasiejowensis, Cyclotosaurus intermedius,* and the phytosaur (*Dzik et al., 2000*; *Dzik, 2003b*; *Dzik & Sulej, 2007*). Remains of terrestrial animals are less frequent, although common (e.g., *Stagonolepis olenkae, Silesaurus opolensis*) (*Dzik et al., 2000*; *Dzik, 2003b*; *Dzik & Sulej, 2007*). According to *Bodzioch & Kowal-Linka (2012)*, the lower bone level was deposited by a single short-lived, high energy event, probably a flood, which is contradicted by strictly horizontal intercalations of grainstone.

The upper bone horizon (a few meters above the lower one) is a lenticular red mudstone body within fluvial deposits. In contrast to the lower bone horizon, terrestrial vertebrates dominate there, whereas aquatic ones are rare (*Dzik, 2003b*; *Dzik & Sulej, 2007*).

## MATERIAL AND METHODS

Four specimens with elements of forearms, carpus and hand preserved in association as well as several various isolated forelimb bones were examined. All humeri used in this study were found isolated. The studied material comes from both bone bearing horizons at the Krasiejów locality. It is a part of the collection of the Institute of Paleobiology, Polish Academy of Sciences in Warsaw. A complete list of specimens with the description, measurements, photographs and taphonomical notes are provided in Appendix S1–S3 and Figs. S1–S9.

Specimens were recognized as representing aetosaurs based on the general morphology of the long bones and the relative proportions between the different elements, the presence of a well-defined humeral head, with well-developed deltopectoral crest, the presence of a long olecranon process in ulna, the presence of fused radiale and intermedium in the carpus, the massive and thick digits, and the presence of osteoderms. Isolated bone elements were identified based on comparison with specimens preserved in association, if possible. All were referred to *S. olenkae,* as there is only one species of aetosaur described thus far from Krasiejów.

The bones of *S. olenkae* are more heavily built and stouter compared to those of rauisuchian *Polonosuchus silesiacus* and phytosaur *Paleorhinus* (=*Parasuchus*) sp., with which they may be confused. The humeri of aetosaurs in comparison to the humeri of phytosaurs have more transversely expanded proximal head than in phytosaurs (the ratio between the entire length of bone and width of the head in aetosaurs is around 0.53 (based on the humeri ZPAL AbIII/1175, 2369, 2627) and in phytosaurs is around 0.35 (based on the humeri ZPAL AbIII/1994, 502/38)). In addition, the deltopectoral crest is more developed in aetosaurs and significantly expanded laterally, while in phytosaurs the lateral margin of the shaft is straight (*Nesbitt, 2011*; *Parker, 2012*; ZPAL AbIII/1994, 502/38). The ulnae of aetosaurs in comparison with phytosaurs (e.g., ZPAL AbIII/3362) have more

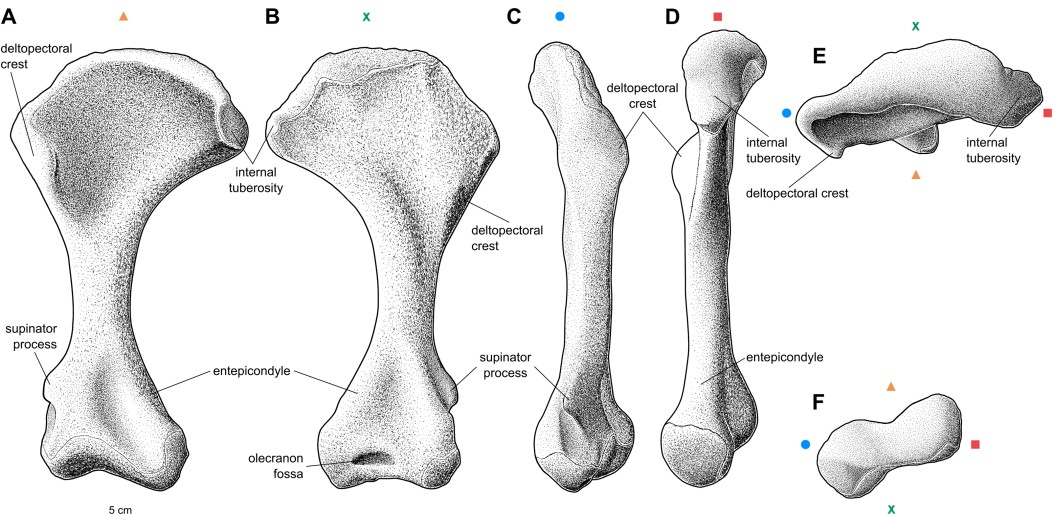

**Figure 1** Reconstruction of the right humerus of the aetosaur *Stagonolepis olenkae*, *Sulej, 2010*.
(A) Ventral view. (B) Dorsal view. (C) Lateral view. (D) Medial view. (E) Proximal view. (F) Distal view.
Symbols attached to pictures show which surface is exposed in the drawing, with (**X**) for the dorsal, (▲)
for the ventral, (■) for the medial, and (●) for the lateral, and how the surfaces are oriented in proximal
and distal view. Scale based on spec. ZPAL AbIII/1175.

expanded proximal articulation surface with pronounced coronoid process (*Nesbitt, 2011*; *Parker, 2012*).

All specimens were cleaned from the surrounding sediment mechanically (with a pneumatic airscribe, PaleoTools model ME-9100) and chemically (with 5% formic acid). Three dimensional models of five specimens were obtained utilizing photogrammetry method with use of several freeware programs (FastStone Photo Resizer, VisualSFM, MeshLab, Texture Stitcher, and DazStudio) and then implemented into .PDF files with Adobe Acrobat (*Cignoni et al., 2008*; *Chuang et al., 2009*; *Furukawa & Ponce, 2010*; *Kazhdan & Hoppe, 2013*; http://ccwu.me/vsfm/). Detailed description of the process and 3D PDFs are provided in the Appendix S4–S9.

## RESULTS

### Osteological description of the forelimbs of the aetosaur *S. olenkae*
#### Proportions and general description of the forelimb bones
Forelimb elements of *S. olenkae* (humerus, Fig. 1; ulna and radius, Figs. 2–4; the largest carpal bone—fused radiale and intermedium, Fig. 5; manus, Fig. 6) are smaller than the corresponding elements of the hind limbs, therefore the entire forelimb must have been shorter than hind limb (D Dróżdż, pers. obs., 2018; *Książkiewicz, 2014*). The humerus is about two-thirds the length of the femur (*Książkiewicz, 2014*). The length ratio of the humerus ZPAL AbIII/2369 to the femur ZPAL AbIII/691 is 0.69 (the specimens possibly belong to a single animal, because they have been found close to each other in the same assemblage). Although there are no humeri found in articulation with radius and ulna, it can be assumed that humerus in *S. olenkae* is longer than each of the forearm bones,

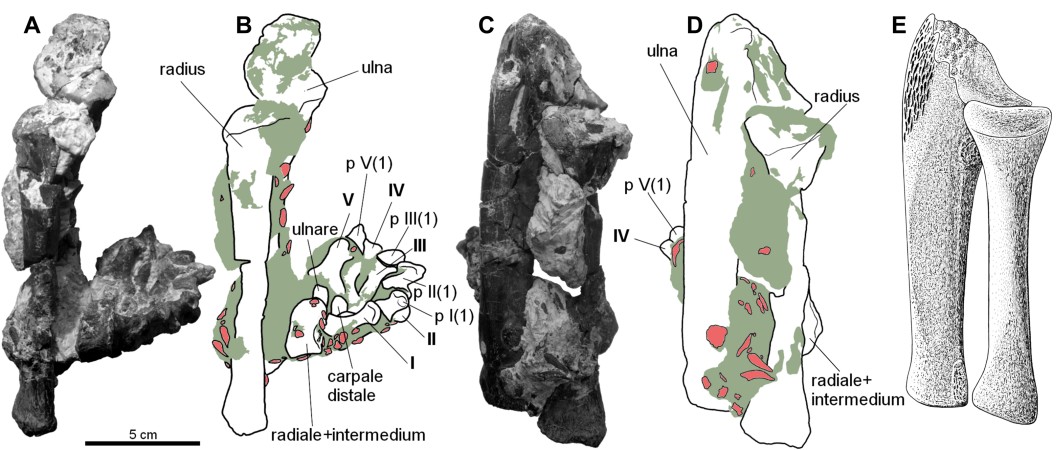

**Figure 2** **Forearm of the aetosaur *Stagonolepis olenkae*, *Sulej, 2010*.** (A) Photograph and (B) schematic drawing of the specimen ZPAL AbIII/2407 with the forearm bones in medial view and (C, D) in dorsal view. (E) Reconstruction of the forearm bones in dorsal view. In the schematic drawings, osteoderms are marked red and the sediment is marked grey. All photographs and drawings are in the same scale.

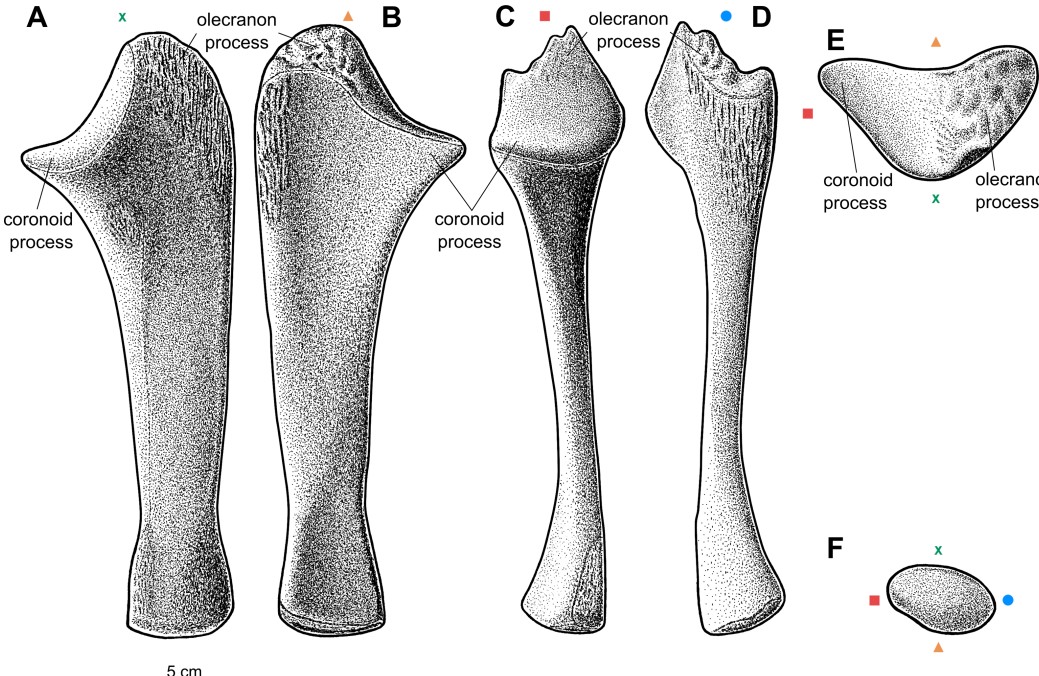

**Figure 3** **Reconstruction of the left ulna of the aetosaur *Stagonolepis olenkae*, *Sulej, 2010*.** (A) Dorsal view. (B) Ventral view. (C) Medial view. (D) Lateral view. (E) Proximal view. (F) Distal view. Symbols attached to pictures show which surface is exposed in the drawing, with (**X**) for the dorsal, (▲) for the ventral, (■) for the medial, and (●) for the lateral, and how the surfaces are oriented in proximal and distal view. Scale based on spec. ZPAL AbIII/3351.

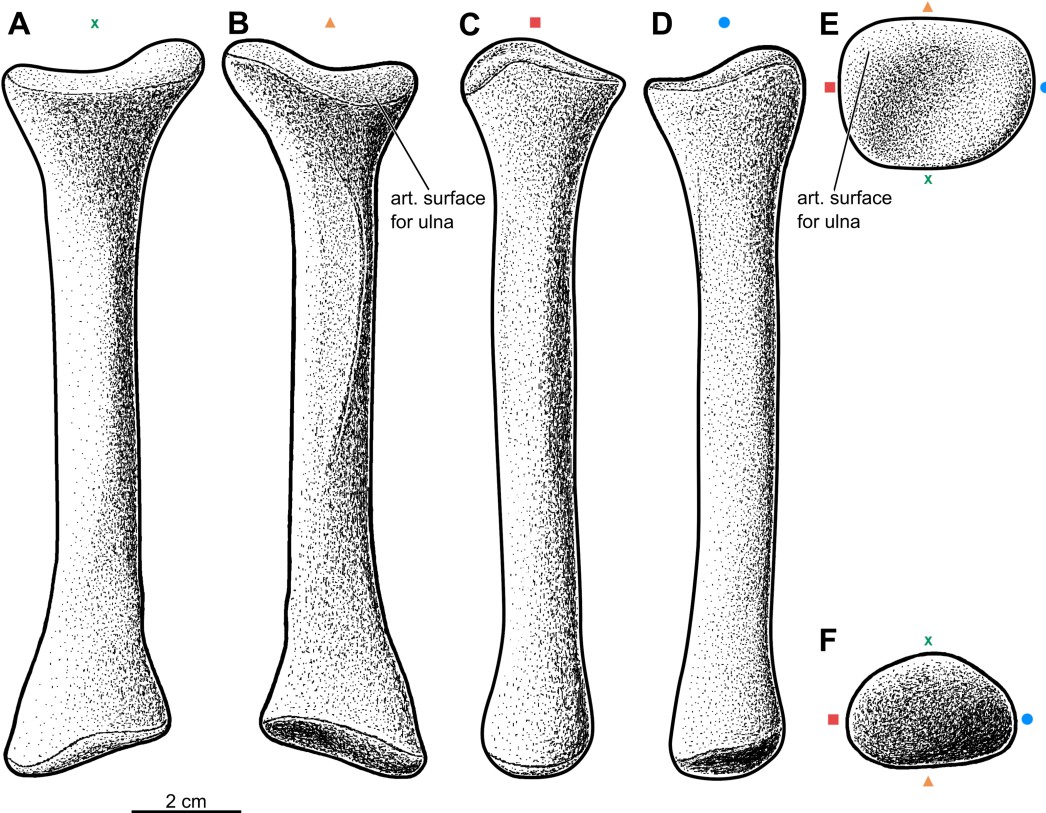

**Figure 4** **Reconstruction of the left radius of the aetosaur *Stagonolepis olenkae, Sulej, 2010*.** (A) Dorsal view. (B) Ventral view. (C) Medial view. (D) Lateral view. (E) Proximal view. (F) Distal view. Symbols attached to pictures show which surface is exposed in the drawing, with (**X**) for the dorsal, (▲) for the ventral, (■) for the medial, and (●) for the lateral, and how the surfaces are oriented in proximal and distal view. Scale based on spec. ZPAL AbIII/3322.

based on the comparison of several specimens (D Dróżdż, pers. obs., 2018; *Ksiażkiewicz, 2014*). Possibly the ulna ZPAL AbIII/1179 belongs to the same animal as the humerus ZPAL AbIII/2369, because of their similar taphonomic condition and relatively close position in the sediment at the moment of recovery. The length ratio of the ulna ZPAL AbIII/1179 to the humerus ZPAL AbIII/2369 is 0.79. The radius and ulna are orientated parallel to each other (Fig. 2; based on specimen ZPAL AbIII/2407). The ulna is longer than the radius and more massive. The ratio of length between the radius and the ulna in the specimen ZPAL AbIII/2407 is 0.85. The proximal ends of the radius and ulna form a single articulation surface for the humerus. The joint between the radius and ulna is elongated and crescent-shaped (Figs. 3A, 4B; based on ZPAL AbIII/3351, 3322), which seemingly makes the relative rotation of these two bones impossible. The olecranon process of the ulna is high (Figs. 2 and 3; based on ZPAL AbIII/2407, 2014, 3351), and in large specimens of humeri the olecranon fossa is present (Fig. 1B; ZPAL AbIII/1175, 257). Based on specimen ZPAL AbIII/2407 the manus together with the carpus is shorter than both the ulna and the radius (Figs. 2A–2D). The carpus consist of at least four bone elements. The largest one of

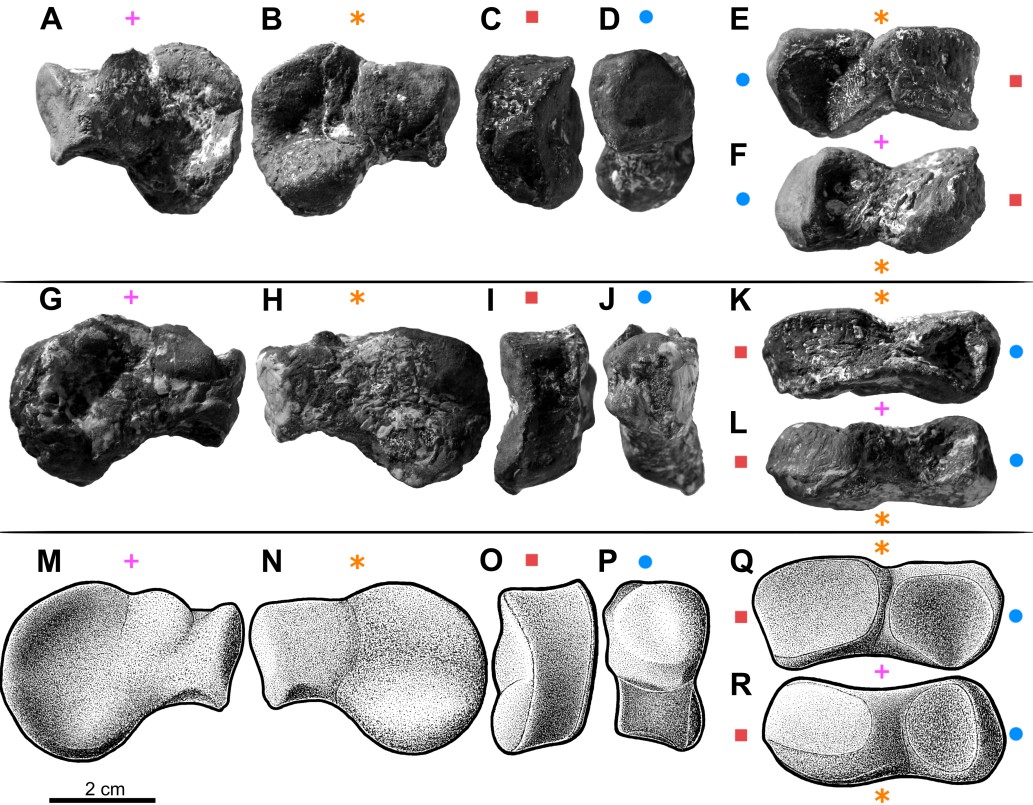

**Figure 5** **The largest carpal bone of the aetosaur** *Stagonolepis olenkae, Sulej, 2010.* Probably fused radiale and intermedium. (A–F) Photographs of the left bone, spec. ZPAL AbIII/2071. (G–L) Photographs of the right bone, spec. ZPAL AbIII/2071. (M–R) Reconstruction of the right bone. (A, G, M) Proximal view. (B, H, N) Distal view. (C, I, O) Medial view. (D, J, P) Lateral view. (E, K, Q) Dorsal view. (F, L, R) Ventral view. Symbols attached to pictures show which surface is exposed in the drawing or photograph, with (+) for the proximal, (*) for the distal, (■) for the medial, and (●) for the lateral, and how the surfaces are oriented in dorsal and ventral view. All photographs and drawings are in the same scale.

them is probably a fused radiale and intermedium (Fig. 5; ZPAL AbIII/2071, 2407, 3349/1, 3349/2). It is connected with the radius and partially with the ulna at the forearm side and the metacarpals I–III and probably IV at palm side. Its arrangement prevents rotation movements of the carpus and restricts moves of the carpal joint only to a one sagittal plane. Metacarpals and phalanges (Fig. 6E) are stout, relatively short and wide. Medial digits (II, III, IV) are almost of equal length and they are noticeably longer than digits I and V, while digit I is longer than digit V (relative length of digits II~III~IV>I>V). Digit I is the most robust in the manus, the medial digits are of similar form and massiveness, while digit V is the tiniest one in the hand. The phalangeal formula is probably 2-3-4-5?-3? and at least digits I to III terminate with claw-like unguals (Fig. 6E, mostly based on ZPAL AbIII/3349/1 and 2071). The ungual of the first digit is the largest one, and it is much bigger than the rest. The ungual of the second digit is about one-third smaller than the first one, and the sizes of following unguals decrease in the same pattern (Fig. 6E; based on ZPAL

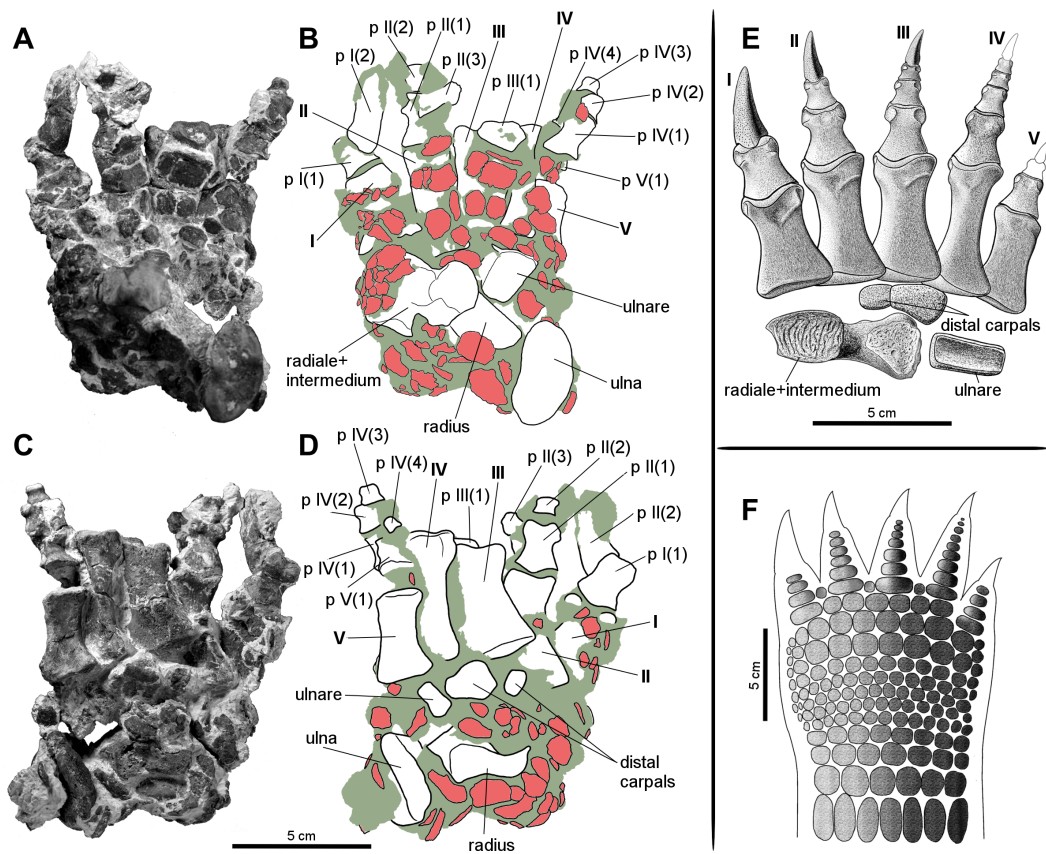

**Figure 6  Manus of the aetosaur *Stagonolepis olenkae*, *Sulej, 2010*.** (A) Photograph and (B) schematic drawing of the right manus spec. ZPAL AbIII/3349/1 in dorsal and (C, D) ventral view. (E) Reconstruction of the right carpus and manus in dorsal view. (F) Hypothetical arrangement of osteoderms in dorsal view. In the schematic drawings osteoderms are marked red and the sediment is marked grey. Photographs and drawings A–D are in the same scale. Restoration (E) of the carpus and manus is based mostly on spec. ZPAL AbIII/3349/1 and 2071.

AbIII/3349/1, 2071). The dorsal part of the hand, forearm and probably also upper arm is entirely covered by osteoderms (Fig. 6F; based on ZPAL AbIII/2407, 3349/1, 3349/2).

### *Humerus*

The humerus of *S. olenkae* (Fig. 1), examined here in the specimen ZPAL AbIII/257, 1175, 2627, is a massive, strongly built bone with a straight shaft and well-defined, transversely wide, proximal and distal head. The shaft is twisted through about 30 degrees, so that the distal end faces backward as well as upward. The proximal head expands into the coronal plane, mostly medially, and is very wide, almost half of the length of the entire bone–the ratio of the medio-lateral width of the humeral head to the entire length of the bone is about 0.54 (Figs. 1A–1B; ZPAL AbIII/1175, 2627). Its articulation surface is convex and it is covered by multiple irregularly arranged tubercles of random size, which indicate the presence of a well-developed cartilaginous cap (ZPAL AbIII/2627). The medial process of the proximal head forms an internal tuberosity (it is almost as robust as the articular

thickening). It is separated from the main articulation surface by a pronounced indentation (ZPAL AbIII/1175, 2627). Below the articulation surface of the proximal head the shaft gently slopes forming a short neck. Under the medial process the neck forms a thin lamina. The deltopectoral crest is well-developed. It starts below the level of the articulation surface of the humeral head and below the level of the internal tuberosity (Figs. 1A–1B; ZPAL AbIII/1175, 2627). The shaft in the medial/lateral view remains straight throughout its entire length (Figs. 1C–1D). In the dorsal/ventral view, the medial edge of the humerus forms an arch (Figs. 1A–1B). The arch is strongly bent, toward the lateral and distal side, in the proximal portion of the shaft, then fluently transits into almost straight line in the middle portion of the shaft, and bends again, toward the medial and distal side, above the medial epicondyle. The shaft expands laterally, in the regions of the deltopectoral crest and the lateral epicondyle, but the expansion is not as significant as on the medial side (Figs. 1A–1B; ZPAL AbIII/1175, 2627). The wide proximal portion of the shaft is much thinner than the middle and distal section. Its thickness increases gradually from the neck of the proximal head, up to the end of the deltopectoral crest (Fig. 1D). On the dorsal surface of the shaft, next to the deltopectoral crest, there is an elevation, which in the specimen ZPAL AbIII/2627 is ornamented by delicate grooves. On the dorsal side, distal to the elevation, close to the medial edge, an elongated knob is present (Fig. 1B). The ventral side of the shaft in the proximal section is smooth (Fig. 1A; ZPAL AbIII/1175, 2627). The middle section of the shaft, below the deltopectoral crest up to the epicondyles, is close to oval in cross-section and of uniform thickness and width. On the dorso-lateral surface there is a straight furrow that originates in the elevated area next to deltopectoral crest and continues up to the epicondyle (Fig. 1C; ZPAL AbIII/1175, 2627). The distal head of the humerus is much narrower than the proximal head, being about 0.6 the length of the proximal head and 0.25 the length of the entire bone. The entepicondyle is wide. The ectepicondylar groove on the lateral side is fully exposed and deep (Figs. 1A–1C; ZPAL AbIII/257, 1175, 2627). The supinator process is thick and prominent (Figs. 1A–1C; ZPAL AbIII/257). In large specimens the olecranon fossa is present (Fig. 1B). In specimen ZPAL AbIII/257 the olecranon fossa is not fully enclosed as in specimen ZPAL AbIII/1175. In distal view, the distal head is transversely elongated, with pronounced narrowing in its center (Fig. 1F). The articulation surface of the distal head is covered by multiple irregularly arranged tubercles of random size, which indicate the presence of well-developed cartilaginous cap (ZPAL AbIII/257, 2627).

### Ulna

The ulna of *S. olenkae* (Figs. 2 and 3) represented by the specimen ZPAL AbIII/2407, 3349/1, 3349/2, 1100/1, 1179, 2014, and 3351, is straight and dorso-ventrally flattened. The proximal portion of the shaft is triangular in cross-section and it is wider and thicker than the middle and distal portion. The olecranon process is long (around 0.2 of the total ulna length in the specimen ZPAL AbIII/2407 and 3351; Figs. 3A–3D), but it was probably even longer as on its top there is a tubercular area (Fig. 3B, 3D–3E), which indicates the presence of well-developed apical cartilage (ZPAL AbIII/2407, 2014, 3351). The tubercles are of different size and are irregularly arranged. The articular surface for the humerus

falls rapidly behind the tubercular area of olecranon process towards the coronoid process, where it becomes almost flat (Figs. 3A–3B; ZPAL AbIII/2407, 2014, 3351). The coronoid process is elongated medially and prominent. The articular surface for the radius (on the dorsal side, below the coronoid process) is elongated and crescent-shaped (Fig. 3A; ZPAL AbIII/2014, 3351). The area below the olecranon process is covered by an irregularly arranged and strongly marked series of pits and grooves (Figs. 2E, 3A–3B, 3D; ZPAL AbIII/2014, 2407, 3351). In crocodiles, this area is the insertion point for the triceps muscle (*Meers, 2003*). The presence of strongly marked structures in this region suggests that the triceps muscle must have been well-developed in *S. olenkae*. There is another small ornamented area (with an ornamentation of small pits and grooves) on the dorsal side of the ulna, in the proximal part of the shaft, distal to the articular surface for the radius (Figs. 2E, 3A; ZPAL AbIII/2014). Its ornamentation consists of small pits and grooves. The shaft in the middle section forms two parallel, almost flat surfaces (on the dorsal and ventral side) and (in dorsal/ventral view) slightly tapers symmetrically towards the distal end of the ulna, where a delicate neck can be distinguished (Figs. 3A–3B; ZPAL AbIII/1100/1, 1179, 2014, 2407, 3351). The lateral edge of the shaft in the middle section is smooth and rounded (Fig. 3D). The medial edge of the shaft, in the middle section, ends with a furrow that initiates below the articular surface for the humerus and the radius, continues throughout the middle portion of the shaft and weakens towards the distal end of the ulna (Fig. 3C). The cross-section of the shaft, in the upper part of the middle section, is similar to an irregular pentagon with two parallel sides (dorsal and ventral surfaces of ulna) and sharp edge oriented medially (where the furrow is). The shape of the cross-section changes towards the distal end of the ulna and becomes more oval. On the dorsal side of the ulna, in the middle section of the shaft, two other well marked longitudinal furrows can be recognized (Figs. 2E, 3A). The first furrow on the dorsal surface, situated next to the lateral edge of the bone, continues almost through the entire length of the shaft and is arched laterally in dorsal view. The other one, situated next to the medial edge of the bone, is straight and continues only through the middle section of the shaft. The ventral surface of the ulna forms a single flat plane with a small longitudinal depression in the middle (Fig. 3B). The plane continues through the proximal and the middle section of the shaft. In the distal part, distal to the delicate neck, the shaft slightly expands and gently twists (Figs. 3C–3D). In distal view, the shape of the surface for the wrist bones is oval (Fig. 3F; ZPAL AbIII/2407, 3349/1). Similar to the proximal end, the distal end of the ulna was also partially cartilaginous. Its bony surface is covered by irregularly distributed tubercles, although much smaller than those in the region of the olecranon process (ZPAL AbIII/2407, 3349/1, 3351). Distal to the neck on the ventro-medial side, there is a prominent oval ornamented area (Figs. 3A, 3C; ZPAL AbIII/3351). Its ornamentation consists of small, irregularly arranged pits and grooves.

### Radius

The radius (Fig. 4) represented by the specimen ZPAL AbIII/1628, 2407, 2016/2, 2016/4, 3322, 3349/1, 3349/2 is shorter than the ulna and longer than the manus. The ratio of length between the radius and the ulna in the specimen ZPAL AbIII/2407 is 0.85 (Figs. 2A–2D). The shaft is straight and of almost uniform thickness throughout its entire length, except

for the proximal or distal ends, where it expands in a funnel-like manner (Figs. 4A–4B). In dorsal-ventral view the proximal and distal ends are of the same width (Figs. 4A–4B; ZPAL AbIII/3322). The articulation surface for the humerus is orientated perpendicular to the shaft (Figs. 4A–4B) and in proximal view it is semi-rounded (Fig. 4E). Its surface is almost flat, with a tiny depression in the centre and a small process at the lateral edge. The articulation surface for the ulna is crescent-shaped and extends longitudinally (Fig. 4B, E; ZPAL AbIII/3322). The middle portion of the shaft is semi-square in cross-section. Two corresponding, sharp, well-marked furrows are present on the dorsal and the ventral side of the shaft (Figs. 4A–4B; ZPAL AbIII/3322). Both furrows are s-shaped, which gives the impression that the shaft is twisted helically. Two furrows of similar pattern, but much less distinct, are also present on the lateral and the medial side of the shaft (Figs. 4C–4D). The bony surface of the articulation area with the carpus is semicircular in distal view (Fig. 4F; ZPAL AbIII/2106/2, 2407, 3322). Its surface is covered by irregularly arranged tubercles (ZPAL AbIII/2106/2), similar to those present on the olecranon process of the ulna and the humeral heads, but smaller. The presence of the tubercles indicates a well-developed cartilaginous finish.

### Carpus

The carpus consists of at least four bony elements, probably oriented in two rows (Fig. 6E). The largest carpal bone is the fused radiale and intermedium (Fig. 5), present in the spec. ZPAL AbIII/2407, 3349/1, 3349/2, 2071, which probably forms the proximal row with the narrow, square-shaped ulnare, present in the specimen ZPAL AbIII/2407, 3349/1, 3349/2, and 2071. The second row consist of at least two bony distal carpals (present in the specimen ZPAL AbIII/2071, 2407 (only one), 3349/1, and 3349/2), among which one is larger and crescent-shaped and the second one much smaller and pea-shaped.

The fused radiale and intermedium (Fig. 5) is a thick bone, elongated in the coronal plane. Proximally it articulates with the radius and partially the ulna, and is adjacent distally to metacarpals I, II, III, and partially IV (Fig. 6; based on ZPAL AbIII/2407, 3349/1). *Książkiewicz (2014)* described a single specimen (UOBS 02609), in which the radiale and the intermedium are separated, but he suggested that the bone is probably broken. The radiale and intermedium are joined together by a suture (Figs. 5M–5N, 5R–5S; ZPAL AbIII/2071, 3349/1). The suture is less pronounced, or totally fades, in the ventral and middle portion of the bone, which suggests a continuous process of fusion of both elements. It seems that the two bones initially ossified separately and fused later in ontogeny. For the purpose of further description, the suture will be used as reference point dividing the bone in two sections: the one corresponding to the radiale and the second corresponding to the intermedium. In proximal-distal view the section corresponding to the radiale is semi-oval or rounded and the section corresponding to the intermedium is rectangular (Figs. 5M–5N). In this views, the section corresponding to the radiale is much larger than the section corresponding to intermedium. In dorsal-ventral view the shape of the fused radiale and intermedium is close to rectangular (Figs. 5Q–5R). On the proximal side of the bone, at the section corresponding to the radiale, the surface forms a prominent ridge surrounding the center of the section dorsally and medially (Fig. 5M;

ZPAL AbIII/2071). Farther, on the proximal side at the section corresponding to the intermedium, a modest but distinct elevation is present (Fig. 5M). The elevation originates at the middle of the proximal articulation surface, and continues towards the dorsal edge of the bone forming a process-like structure. On the distal side, at the section corresponding to the radiale, there is a large, prominent tuber (Figs. 5N–5O; ZPAL AbIII/2071). The tuber originates in the middle of the articulation surface, and continues towards the distal edge, covering a little less than half of the section corresponding to the radiale (Fig. 5N). Excluding the tuber region, the articulation surface of the distal side of the fused radiale and intermedium forms an almost uniform plane, with a modest elevation close to the dorso-medial edge and in the section corresponding to the intermedium (Fig. 5N; ZPAL AbIII/2071). On the dorsal surface of the fused radiale and intermedium, two distinct areas are visible, one at the section corresponding to the radiale, and one at the section corresponding to intermedium (Figs. 5Q, 6C; ZPAL AbIII/2071, 3349/1). The area in the section corresponding to the radiale is flat with a small depression in its central part. It continues through the medial and part of the ventral side of the section. The area in the section corresponding to the intermedium forms a deep depression. Opposite to it, on the ventral side of the section corresponding to the intermedium, there is another area that forms depression but it is smaller and shallower (Fig. 5R; ZPAL AbIII/2071). The lateral side of the fused radiale and intermedium forms one square-shaped articulation area with a prominent tuber in the distal-dorsal corner (Fig. 5P).

The ulnare is a cube with square bases and rectangular sides. The width of the sides is half the width of the bases. It was probably arranged in one line with the fused radiale and intermedium, with the bases oriented proximally/distally (based on the specimen ZPAL AbIII/2407, 3349/1). The bases are almost flat, with slight depressions in their centres.

The carpus consists of at least two other (probably distal) carpals. One of them (larger) is elongated and lunar-shaped, and the other one is about five times smaller and pea-shaped. These are probably distal carpals III and IV, and based on the specimens ZPAL AbIII/2407 and 3349/1 they are arranged adjacent to metacarpal III in the second (distal) row of carpals.

The elongated lunar-shaped bone in specimen ZPAL AbIII/2071, alternatively may be interpreted as a pisiform, based on the comparison with modern scratch-digging mammals, such as badger (*Hildebrand, 1988*). If true, it should be oriented in one line with ulnare. Presence of long pisiform is characteristic for digging mammals (*Hildebrand, 1988*).

### Metacarpals

The metacarpals (Fig. 6E) present in the specimen ZPAL AbIII/2071, 2102, 2407, 3349/1, 3349/2 and are robust, relatively short, wide and dorso-ventrally flattened. Their bases are wider than the heads. Their shafts taper towards the distal ends. They match and partially cover each other in dorsal view. Metacarpals II, III, and IV are of similar shape and length, and are noticeably longer than metacarpals I and V, which are about 0.75 their size (based on ZPAL AbIII/2071, 2407, 3349/1, 3349/2). Metacarpal I is slightly longer than metacarpal V. The relative length between the metacarpals I, II, and III vary among specimens. For example, in specimen ZPAL AbIII/3349/1, metacarpal IV is the

longest, but in specimen ZPAL AbIII/2407 the longest is metacarpal III. Metacarpal I is the most robust, and metacarpal V the most gracile. The relationship of robustness can be described as I<II<III<IV<V. The distal articulation surfaces of the metacarpals are slightly asymmetric, each with a larger tuber on the medial side. Metacarpal I is also the widest among the metacarpals (Fig. 6E; ZPAL AbIII/2071, 2407, 3349/1, 3349/2). Its shaft is flat dorsally and rectangular in cross-section. On the ventral surface it has a depression for the subsequent metacarpal. Metacarpals II, III, and IV are of similar shape (Fig. 6E; spec. ZPAL AbIII/2071, 2102, 3349/1, 3349/2). Their shafts are triangular in cross-section. Like in metacarpal I, there is a depression on their dorsal surface for the subsequent metacarpal. The shape of most gracile metacarpal V differs between specimens. In specimen ZPAL AbIII/3349/1 and 3349/2 it is wide and flat. Its width is almost uniform throughout the entire length, and the base and the head are not distinct. In the specimens ZPAL AbIII/2071 and 2407 the base and the head are much wider than the shaft and well-developed. The shaft is slender and oval in cross-section.

### Phalanges

The phalanges (Fig. 6E) are present in the specimen ZPAL AbIII/2407 (Figs. 2A–2D)—all phalanges of the first row, ZPAL AbIII/3349/1 (Figs. 6A–6D)—all phalanges of the first and second digit, broken 1st row phalanx of the third digit, the four phalanges of the fourth digit and 1st row phalanx of the fifth digit, ZPAL AbIII/3349/2—all phalanges of the first row, ZPAL AbIII/2071—in the right hand all phalanges of the first, second and third digit, in the left all phalanges of the first and second digit, and ZPAL AbIII/257, 3352, 3353—isolated phalanges. The probable phalangeal formula for *S. olenkae* is 2-3-4-5?-3? (Fig. 6E), based mostly on the specimens ZPAL AbIII/3349/1 and 2071. The number of the phalanges for the first three digits is certain, because they terminate with claw-like unguals preserved in the first two digits of ZPAL AbIII/3349/1 (Figs. 6A–6D), and first three digits of the right manus and the first two of the left of the spec. ZPAL AbIII/2071. In ZPAL AbIII/3349/1 the phalanx of the fourth row of the digit IV ends with an articulation surface, which indicates the presence of another phalanx or an ungual. However because of small size of the preserved fourth phalanx, it is unlikely that there was more than one element following it (Figs. 6C–6D). As for the fifth digit, the size of the first row phalanx, compared to the size of other phalanges in ZPAL AbIII/3349/1, suggest that at least two phalanges and an ungual were present. In UOBS 02834 described by *Ksiażkiewicz (2014)*, containing hand elements preserved in articulation, two phalanges of the fifth digit are present. The phalanges are dorso-ventrally flattened, short, and wide (Fig. 6E). The phalanx base is always wider than its head. The shaft narrows towards the distal end. It is rounded at the dorsal side and flat at the ventral side. Both the base and the head are slightly asymmetric. In the heads the tuber on the internal side is always larger than that on the external side and a depression is present on the articulation surface. This feature is more pronounced in the phalanges closer to the metacarpals. Interlocking phalangeal articular surfaces are wide and they extend deep into the shaft. The grooves for ligament attachments are well marked in all phalanges despite their size (Fig. 6E).

As mentioned above, claw-like unguals are present on at least the first three digits (Fig. 6E; based on ZPAL AbIII/2071, 3349/1). The size of the claw-like unguals decreases in more lateral digits. The largest claw-like ungual of the first digit is about one-third longer and more massive than that of the second digit, and the second is longer and more massive than the third one in the same manner (ZPAL AbIII/2071). It can be inferred from the size of the preserved phalanges that other claw-like unguals (if present) kept this tendency. The unguals are laterally compressed, with sharp edges at the top and bottom sides (similar to claws of, for example, armadillos, pangolins, badgers) (*Hildebrand, 1988*). They are tear-shaped in cross section but asymmetric, with a depression on the medial surface (ZPAL AbIII/2071). Longitudinal grooves for ligaments are well-marked on both lateral and medial surfaces. Almost the entire surface of the unguals is covered by tiny and very densely distributed perforations (Figs. 6A–6B, 6E; ZPAL AbIII/2071, 3349/1). Similar texture can be observed on the bony parts of horns, for example, in modern bovids or in the unguals of armadillos (personal observation, *Hildebrand, 1983*). It indicates the presence of a well-developed keratin sheath. Considering the general morphology of the whole hand it is probable that very small claw-like unguals were present on the fourth and the fifth digits. In some species of modern digging animals such as armadillos, pangolins, or moles often one or several digits enlarge and take a blade-like shape useful for a scratch-digging, while the others are considerably smaller, reduced or absent (*Beddard, 1902*; *Hildebrand, 1988*).

### Dermal skeleton

Probably the entire forearm of *S. olenkae* was covered by numerous appendicular osteoderms (Figs. 6A–6B, 6F). They are preserved in association with the arm, carpus and manus elements in ZPAL AbIII/2071, 2407, 3349/1 and 3349/2. In the specimens ZPAL AbIII/2407, 3349/1, and 3349/2 they are accumulated mostly on the dorsal side of the hand. In the spec. ZPAL AbIII/2407 there is also a large cluster of osteoderms, previously recovered in front of the distal end of radius and ulna, but removed during preparation (not illustrated). The appendicular osteoderms are generally flat, plate-like structures, semi-round to semi-oval in dorsal/ventral view. The edges of the osteoderms can be regular and smooth (mostly in the larger scutes) or irregular and ridged (more often in smaller ones). They are of various sizes, the largest are about 2.5 cm in diameter (ZPAL AbIII/2407, separated cluster of osteoderms), the smallest around 0.5 cm in diameter (several osteoderms in ZPAL AbIII/2407, 3349/1, and 3349/2). They are ornamented on the dorsal surface. The ornamentation consists of delicate grooves and depressions. The ventral surfaces of the appendicular osteoderms are smooth. On both dorsal and ventral side of osteoderms, tiny openings for blood vessels are present.

The appendicular osteoderms in ZPAL AbIII/2407, 3349/1 and 3349/2 are significantly displaced in regard to their *in vivo* position, likely because of transportation and early diagenesis processes. However, some general observation can be made. The appendicular osteoderms cover the entire dorsal surface of the carpus and the manus (Fig. 6F). The osteoderms that occur in this area are of various sizes and shapes, but generally they are semi-round and small to medium (with diameter about 0.5 to 1.5 cm). In more flexible

regions, such as the carpus and joints of the digits, the osteoderms are smaller, but in greater number than in more static regions, such as above the metacarpal shafts, where they are larger but less numerous. As for the region of the arm, it is likely that it was entirely covered by appendicular osteoderms in the manner restored for *Aetosaurs ferratus* or *T. coccinarum* (*Schoch, 2007*; *Heckert et al., 2010*). The osteoderms of the arm are larger (up to around 2.5 cm in diameter). The lack of articulation structures on the surfaces of the appendicular osteoderms suggests that they did not overlap with each other (unlike the rectangular osteoderms presents on the back of the animals), but rather lay one next to another like the scutes of modern crocodiles and alligators (e.g., *Grigg & Gans, 1993*).

## DISCUSSION

### Comparisons with other aetosaurs

The forelimbs of *S. olenkae* are robust, with the humerus having a prominent wide head and condyles, and well-developed deltopectoral crest, dorso-ventrally flattened ulna with high olecranon process, straight stout radius, and sturdy short digits ending with claw-like unguals, with flattened, wide metacarpals and phalanges. Based on the literature they are much more similar to other large and middle size aetosaurs like *S. robertsoni*, *L. meadei*, *T. coccinarum*, *Typothorax antiquum*, *Desmatosuchus smalli*, *Desmatosuchus spurensis*, or *Neoaetosauroides engaeus* (*Sawin, 1947*; *Walker, 1961*; *Bonaparte, 1971*; BJ Small, 1985, unpublished data; *Long & Murry, 1995*; *Heckert & Lucas, 2002*; JW Martz, 2002, unpublished data; *Parker, 2008*; *Heckert et al., 2010*; *Desojo et al., 2013*) and differ from the small ones like *Polesinesuchus aureoli* and *A. ferratus*, in which the above features are less developed (*Roberto-Da-Silva et al., 2014*; *Schoch, 2007*). However *S. olenkae* has also unique features in forelimbs that distinguish it from other species in which they are known, namely: (1) presence of dermal amour covering the dorsal part of the manus, (2) presence of enlarged ungual of the first digit in regard to other unguals of the manus, (3) and in having square cross-section of the radius. Detailed comparisons of *S. olenkae* with other aetosaurs as well as notes on their variability are given in Appendix S10 and Figs. S10–S12.

### Comment on *S. robertsoni* and *S. olenkae*

I personally studied *S. robertsoni* material and I think that differences in the postcranial skeleton between the *S. olenkae* and *S. robertsoni* are generally minor, which was first pointed out by *Lucas, Spielmann & Hunt (2007)*. Nevertheless I need to mention that material of *S. robertsoni*, which consist mostly of casts and incomplete specimens, is often difficult to directly compare with *S. olenkae*, as the specimens often do not have or not have enough preserved essential parts that could be used in comparisons. On the other hand, *Książkiewicz (2014)* specified list of differences in post-cranium between those two species, but does not include any forelimbs characters. Referring to the description of *S. robertsoni* provided by *Walker (1961)*, I have found another three characters (in spite of those three unique for *S. olenkae*) that may distinguish those two species, namely: (1) the presence of an indentation on the humeral head that separates the internal tuberosity and main articulation surface in *S. olenkae*, (2) a more transversely expanded humeral head in

*S. olenkae* (pointed out also by *Parker, 2016*; *Parker, 2018*), (3) and a sharp ended coronoid process in *S. olenkae*. It must be, nonetheless, remembered that *Walker (1961)* also based his interpretations on the same incomplete and poorly preserved material, and thus some of his restorations may be speculative or inaccurate.

### Probable scratch-digging behavior in *S. olenkae*

*S. olenkae* has several features present in modern scratch-diggers according to *Hildebrand (1983)*, *Hildebrand (1988)* and *Coombs (1983)* which are explained in detail in the supplementary material (Appendix S11). Without a doubt, it was able to produce great out-forces with its forelimbs. They are: (1) of strong, robust build (2) with short radius (shorter than the humerus), (3) have a short carpus, with short, stout metacarpals, and (4) have a short, broad phalanges. (5) The deltopectoral crest of the humerus (attachment of deltoid muscles) is well-developed and (6) spans almost half the length of the bone. In the distal part of the humerus, (7) the medial entepicondyle is prominent and wide (comparable to that of armadillos, pangolins, anteaters, and aardvarks; (*Hildebrand, 1983*), and on the lateral side (8) a distinct supinator process is present. (9) The olecranon process of the ulna is long (minimum 0.2 of the total ulna length—the level of scratch-digging ground squirrels (*Hildebrand, 1988*), but for sure it was longer in *S. olenkae*, because of cartilaginous expansion), and has strong and well-marked insertions of the triceps muscle. Following the criteria used by *Heckert et al. (2010)* in regard to *T. coccinarum*, *S. olenkae* bears nine of the eleven characters associated with digging behavior.

Furthermore, the joints of *S. olenkae* are modified for stabilizing the arm. The wrist joint is hinge-like due to the presence of the fused radiale and intermedium. The radius and the ulna are oriented parallel to each other and are immovable relative to each other in the elbow joint. The head of the humerus is very expanded medio-laterally which suggest that it had higher mobility in the horizontal than in the vertical plane. In the autopodium, the joints between the metacarpals and the phalanges are almost flat-ended, the joints between phalanges are slightly V-shaped, and the surfaces of more proximal phalanges are not larger than those of the distal ones. The second-to-last phalanges in the first, second, and third digits have enlarged distal ends with a great curvature radius. The autopodial bones of *S. olenkae* did not have bony stops and I do not recognize sesamoids, however the whole hand was covered by osteoderms that might stiffen it. In modern crocodiles, dorsal osteoderms together with dorsal muscles form a complex structure that strengthens the vertebral column (Frey, 1988). The small number of free phalanges (one in the first digit and two in the second digit), obviously helped in the digging process, however, it is a typical condition in crocodiles and among many basal archosaurs (for example *Postosuchus*, *Parasuchus*, *Riojasuchus*) (*Bonaparte, 1971*; *Szarski et al., 1976*; *Chatterjee, 1978*; *Peyer et al., 2008*; *Weinbaum, 2013*).

The manus of *S. olenkae* was also adapted for breaking compacted soil. Its unguals are laterally compressed, claw-like, and elongated in the manner of modern armadillos, pangolins, echidnas, or moles (*Hildebrand, 1983*). Their surface ornamentation is similar to that on the unguals of, for example, armadillos (*Hildebrand, 1983*), or on the bony surface of Bovidae horns (personal observation), which indicates the presence of a strong

keratin sheath. In addition, the ungual of the first digit in *S. olenkae* is enlarged in respect to the others, similar to what is found in some pangolins and armadillos in which the primary digging digit become enlarged (*Beddard, 1902*; *Hildebrand, 1983*; *Hildebrand, 1988*; *Gaudin, Emry & Morris, 2016*).

## Implications for mode of life in *S. olenkae*

*Sulej (2010)*, based on the presence of a skull endocast with a very large olfactory tract and bulbous, large nares, and the shape of teeth, considered *S. olenkae* to be an omnivorous animal relaying mostly on smell in its search for food and using its shovel-like snout to extract invertebrates and plants from under the ground, and comparing its lifestyle to modern wild boars. Considering several adaptations for scratch-digging recognized here for *S. olenkae*, it safe to assume that it probably started digging with its forelimbs to break and loosen the soil, and then proceeded with its snout. Modern scratch-diggers (such as armadillos, pangolins, and aardvarks) can balance their body with hind limbs and long tail to apply additional strength for a breaking strike (*Hildebrand, 1988*), which is also expected in *S. olenkae*, due to its relatively large size and long tail (a feature characteristic of aetosaurs, *Desojo et al., 2013*, described also for *S. olenkae* by *Ksiażkiewicz, 2014*).

The presence of specialized claws, together with the dermal armor covering the entire forelimb suggests an analogy to the insectivorous mode of life of armadillos or pangolins, considered by some authors to be modern analogues of aetosaurs (e.g., *Bonaparte, 1971*; *Small, 2002*; *Desojo et al., 2013*). The armor protects them against the insects (ants, termites or beetles) they mostly feed on, as well as larger predators they cannot outrun (for example leopards or hyena in the case of pangolins) (e.g., *Talmage & Buchanan, 1954*; *Deligne, Quennedey & Blum, 1981*; *Yang et al., 2013*; *Wang et al., 2016*). A few beetle elytra have been reported from Krasiejów (*Dzik & Sulej, 2007*) and the diet of the nine-banded armadillo may consist of even 40% of coleopterans (*Talmage & Buchanan, 1954*). Several authors suggest therefore the beetles could be the important source of food for aetosaurs (e.g., *Small, 2002*); however, it is hard to imagine an animal being almost as long as a mid-sized automobile to base its diet mostly on insects. Accordingly, the wild boar analogy seems more appropriate because of *S. olenkae*'s overall size. Besides, the wild boar's omnivorous diet includes roots, tubers, bulbs, nuts, seeds, bark, insects, and other smaller animals, and also includes scavenging, but the majority of its food consists of items dug from the ground (*Heptner et al., 1989*). This covers every kind of feeding behavior that has ever been suggested for aetosaurs (*Desojo et al., 2013*). Enlarged osteoderms forming an extensive dermal carapace are most likely a result of a selective pressure from large predators, rather than protection against small arthropods. The upper size range of aetosaurs is comparable to associated rauisuchids, which possibly hunted them (*Drymala & Bader, 2012*) and in Krasiejów, *S. olenkae* is the only large land animal of a size comparable to the local predator *Polonosuchus silesiacus* (e.g., (*Dzik & Sulej, 2007*). Still, *S. olenkae* is the only known aetosaur with the osteoderms covering the dorsal surface of the hand. As the osteoderms take part in thermoregulation (*Farlow, Hayashi & Tattersall, 2010*) and may buffer lactic acid (*Jackson, Andrade & Abe, 2003*), their presence may support *S. olenkae*'s

warming up or giving up heat generated by the intense work of the forelimb muscles, and help to sustain longer high activity of the forelimbs.

During the deposition of bone-bearing horizons in Krasiejów, the climate was semi-tropical with distinct seasonal wet and dry periods (*Gruszka & Zieliński, 2008*). In modern tropical areas during the dry season deciduous plants (including trees, shrubs, herbaceous) protect themselves against dehydration by losing their foliage, and some of them develop resting underground organs rich in storage polysaccharides and proteins (*Bullock & Solis-Magallanes, 1990*). *S. olenkae* may have temporarily relied on such source of food. A similar climate with distinct seasonal wet and dry periods was also recognized in Upper Triassic Chinle Formation in the USA (e.g., *Dubiel, 1984*; *Dubiel, 1987*; *Simms & Ruffell, 1990*), from which several aetosaur species have been described (e.g., *Desmatosuchus, Typothorax, Paratypothorax, Calyptosuchus, Scutarx*; Long & Ballew, 1985; *Long & Murry, 1995*; *Heckert & Lucas, 2000*; *Desojo et al., 2013*; *Parker, 2016*; *Parker, 2018*).

## Probable scratch-digging in other aetosaurs species

Although the possible ability to dig with forelimbs was previously proposed only for *S. robertsoni* and *T. coccinarum* (*Walker, 1961*; *Heckert et al., 2010*), many features associated with the scratch-digging behavior can also be recognized in the forelimbs of mid-sized and large aetosaurs. Namely: (1) robust forelimb bones, with (2) the humerus having a prominent, well-developed deltopectoral crest, and (3) a wide entepicondyle (*Aetobarbakinoides brasiliensis, Argentinosuchus bonapartei, Desmatosuchus smalli, Desmatosuchus spurensis, Longosuchus meadei, Neoaetosauroides engaeus, T. coccinarum, T. antiquum, S. robertsoni*), (4) marked supinator process (*S. robertsoni*), (5) long and marked olecranon process of ulna (*D. smalli, L. meadei, N. engaeus, T. coccinarum, T. antiquus, S. robertsoni*), (6) short and stout metacarpals, (7) short and broad phalanges (*L. meadei, T. coccinarum, S. robertsoni*), and (8) the fused radiale and intermedium in the carpus (*L. meadei* and *S. robertsoni*) (*Sawin, 1947*; *Walker, 1961*; BJ Small, 1985, unpublished data; *Long & Murry, 1995*; *Lucas, Heckert & Hunt, 2002*; *Heckert & Lucas, 2002*; JW Martz, 2002, unpublished data; *Lucas & Heckert, 2011*; *Desojo, Ezcurra & Kischlat, 2012*). Many of mid-sized and large aetosaurs also have the shovel-like expansion at the end of the snout (e.g., *D. smalli, D. spurensis, L. meadei, N. engaeus, T. coccinarum, S. robertsoni*) (*Sawin, 1947*; *Walker, 1961*; *Small, 2002*; *Parker, 2005*; *Parker, 2008*; *Desojo & Báez, 2007*; *Heckert et al., 2010*; *Desojo et al., 2013*). Therefore, it is probable that many of them could also perform scratch-digging, and had a similar mode of life as *S. olenkae*.

Despite the lack of many characters typical for diggers, the forelimbs of diminutive *A. ferratus* have proportions typical for digging animals, with the radius shorter than the humerus, and the palm shorter than the radius (*Schoch, 2007*). *A. ferratus* also has a prominent supinator process and transversely elongated element in the carpus (radiale), but lacks the shovel expansion in the snout (*Schoch, 2007*). This suggests that it could also perform scratch-digging with the forelimbs (probably even better than contemporary animals), but not to the degree of larger aetosaur species.

## CONCLUSIONS

The forelimbs of *S. olenkae* have a morphology characteristic for large and middle sized aetosaurs (over two m long, such as *Desmatosuchus, Typothorax, Longosuchus*) and differs greatly from the smaller ones (below one m long, such as *Aetosaurus, Polisinesuchus*). *S. olenkae*, also has few features that have not been described thus far for any other aetosaur, namely: (1) presence of dermal armor covering the dorsal part of the manus, (2) the presence of an enlarged ungual of the first digit, (3) and having a square cross section of the radius.

It appears that some of the features observed in the forelimbs of all aetosaurs, like the transverse expansion of the humeral head, transverse expansion of the condyles, elongation of the olecranon process of the ulna and general increase of bone robustness, are associated with the increase of size of certain species. However, to fully understand complexity of this process among aetosaurs, further studies on ontogeny and intraspecific variation are needed.

Six characters that distinguish *S. olenkae* and *S. robertsoni* have been recognized in this study (in addition to three unique for *S. olenkae*): (4) more transversely expanded humeral head in *S. olenkae*, (5) distinct greater trochanter (internal tuberosity) in *S. olenkae*, and (6) sharp ended coronoid process of ulna in *S. olenkae*). Additionally some specimens of *S. olenkae* have an olecranon fossa not present in *S. robertsoni*. These differences may furthermore support the distinction of those two species, therefore supporting the establishment of the species *S. olenkae* for the Polish material. However, this statement needs to be treated with caution, because the intraspecific variation and particularly the ontogeny of both species is not well known yet, and the described material of *S. robertsoni* usually does not allow for the detailed comparisons due to the poor state of preservation.

*S. olenkae* was an effective scratch-digger. It has many adaptations associated with producing great forces against hard substrates, stabilizing the joints, and breaking compacted soils. It probably started digging with its forelimbs to loosen and shatter the earth, and then proceeded with its shovel-expansion at the tip of the snout, in search for food underground.

Characters connected with scratch-digging can be recognized also in forelimbs of other aetosaur species, and in most of them they are associated with the presence of the shovel-like expansion in the skull. Therefore the scratch-digging behavior proposed here for *S. olenkae* was probably widespread among mid- and large-sized aetosaurs.

Possible digging behavior in aetosaurs could have co-evolved in an environment with seasonal dry and wet periods like that present during the deposition of the Krasiejów strata. In such environments, many plants develop underground resting organs of high dietary value, which could serve as a source of energy during drought.

## ACKNOWLEDGEMENTS

I would like to thank my supervisors Tomasz Sulej and Jerzy Dzik, for guidance and help during the research and work on this paper; Tomasz Szczygielski for checking the early version of the text and figures, and advice on how to improve it; Mateusz Tałanda,

Grzegorz Niedźwiedzki, Julia Desojo, and Martin Ezcurra for advice on manuscript and sharing literature with me; Voltaire Neto for sharing with me some photos and information about South American aetosaurs; Michel Kazhdan for helping me with the Texture Stitcher; Stig Walsh, Janet Trythall and all volunteers of Elgin Museum, especially Alison Wright and Dave Longstaff for being great hosts during my visit in Scotland; Krzysztof Książkiewicz, for sharing his doctoral thesis with me and showing me the collection it was based on; Justyna Słowiak, Przemysław Świś, Łukasz Czepiński, and Maciej Pindakiewicz for discussion and technical comments about the research and the paper; William Parker and the second anonymous reviewer for corrections and advice on improvement of the manuscript; and Emilia Sałatkiewicz for advice on improvement of the figures and emotional support, without her encouragement this paper would never have come out.

### Funding
The author received no funding for this work.

### Competing Interests
The author declares there are no competing interests.

### Author Contributions
- Dawid Dróżdż conceived and designed the experiments, performed the experiments, analyzed the data, contributed reagents/materials/analysis tools, prepared figures and/or tables, authored or reviewed drafts of the paper, approved the final draft, prepared the 3D models.

### Data Availability
Raw data are provided in the Supplemental Files. And link to online at http://dx.doi.org/10.7717/peerj.5595#supplemental-information.

### Supplemental Information
Supplemental information for this article can be found online at http://dx.doi.org/10.7717/peerj.5595#supplemental-information.

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
