# Peer review of "Osteology of a forelimb of an aetosaur Stagonolepis olenkae (Archosauria: Pseudosuchia: Aetosauria) from the Krasiejów locality in Poland and its probable adaptations for a scratch-digging behavior"

_PeerJ, doi:10.7717/peerj.5595_

## Round 0.1 · original submission · Minor Revisions

· Academic Editor

Minor Revisions

I agree with our reviewers that your paper is publishable after following all their suggestions. Please, take them into full consideration.

·

Basic reporting

Manuscript is very well written for an author for whom English is not their first language; however, there are still quite a few grammatical errors that I have made suggestions for correcting throughout the text. The background materials are sufficient and literature references appropriate. The article is well-figured and has a good amount of useful supplemental material.

Experimental design

This is not the first time digging has been proposed for aetosaurs, but this is the most thorough treatment of that hypothesis to date. The hypothesis is well supported and discussion is clear. Methods are sufficiently described including in the supplemental materials. I have no suggested improvements.

Validity of the findings

The data are robust and conclusions well stated. I have no suggested improvements.

Additional comments

This is a well designed study and a well organized and written manuscript. There are some grammatical errors that need to be corrected.

Reviewer 2 ·

Basic reporting

The article is quite well-written and easily understood. It would benefit from some editing by a native English speaker, but otherwise there are few problems.

Generally speaking the references are adequate, although I think Long and Murry (1995) should be cited more often and Long and Ballew (1985) would also be good to include.

Personally, I would like to see a Figure 1 with a map and generalized stratigraphic understanding of the locality—I just think it's always good to try to provide geographic and geologic context of the fossils, even if that's not the primary thrust of the paper.

Although the figures themselves are well-executed, they would benefit if they were subdivided (e.g., Fig. 1A, 1B, !C) and if the author would take advantage of that, as it is sometimes difficult to decide which figure to look at to see specific morphology described in the text. I'm also not convinced that the figures are cited consecutively (e.g., Fig. 1 is the first figure called out).

Experimental design

This paper is straightforward and thorough, basically, I think that the scientific content is sound and well worth publishing. All of the criteria Peer J lists for this section (original, well-defined, relevant, rigorous, replicable) are satisfied.

Validity of the findings

I believe the author has adequately documented their work and I think that their findings are valid. I do think that it may be beneficial to cite a reference or two regarding things such as the presence of keratin inferred from bone texture (more than just "personal observation) as there's certainly literature out there.

Additional comments

Overall this was an easy paper to review, and I think it should be published with minor revision. Other than my previous comments on the figures, the only major problem is that there are parts of 3 different sentences (none complete) on line 569.

I do think that a close reading by a native English speaker/editor will take the text from "very good" to "excellent." I am traveling so cannot easily scan my copy of the manuscript, but I will try to find a way to send it to the editors.

Obviously I'm biased, but I think Heckert et al. (2010) should be cited on line 32 and Heckert and Lucas (2000) on lines 36, 92, and 595.

Long and Murry (1995) should be cited on line 83, 595, 608

260—"figure 8"-shaped, perhaps?

271—concaves is not a word, a different word is needed there.

line 595 should include citations of Long and Ballew (1985)

Line 898—italicize _Desmatosuchus_

My copy of Supplementary Figure 12 includes many typos in the captions (mostly missing spaces between words). Also, the correct attribution of species for _Longosuchus_ is _Longosuchus meadei_ (Sawin) 1947, as Sawin named the species, which was reassigned to the new genus _Longosuchus_ by Hunt and Lucas (1990).

Although the limbs are incomplete, the author may want to look at the small aetosaur _Coahomasuchus_. I think that _C. chathamensis_ (Heckert et al., 2017) has enough preserved to at least see what the ratio of the ulna-radius to the humerus is.

Heckert, A. B., Fraser, N. C., and Schneider, V. P., 2017, A new species of Coahomasuchus (Archosauria: Aetosauria) from the Upper Triassic Pekin Formation, Deep River Basin, North Carolina: Journal of Paleontology, v. 91, no. 1, p. 162-178.

See also: Heckert, A.B., and Lucas, S.G., A new aetosaur (Archosauria:Crurotarsi) from the Upper Triassic of Texas and the phylogeny of aetosaurs. Journal of Vertebrate Paleontology, v. 19, p. 50-68. http://www.jstor.org/stable/4523969

Annotated reviews are not available for download in order to protect the identity of reviewers who chose to remain anonymous.

---

## Round 0.2 · accepted · Accept

· Academic Editor

Accept

Thank you for take into consideration the suggestions of our reviewers. I think that the manuscript is ready for publication.

#